# Assessment of Thrips Diversity Associated with Two Olive Varieties (Chemlal & Sigoise), in Northeast Algeria

Randa Mahmoudi [1],*, Malik Laamari [1] and Arturo Goldarazena [2]

1    LATPPAM Research Laboratory, Department of Agriculture, University of Batna 1, Batna 05000, Algeria
2    National Museum of Natural Sciences, National Reference Laboratory for Nematodes and Arthropods,
     Department of Biodiversity and Evolutionary Ecology, Calle Serrano 115, 28006 Madrid, Spain
*    Correspondence: randa.mahmoudi@univ-batna.dz

**Abstract:** In this study, the diversity of thrips (Insecta: Thysanoptera) on two varieties of olive trees (Chemlal and Sigoise) in northeast Algeria (Province of Batna), was evaluated for 3 years (2019–2021). In addition, the fluctuations in the numbers of phytophagous thrips were estimated according to the varieties phenological stages. A total of 19 species are identified and the olive thrips (*Liothrips oleae*) have just been reported for the first time in Algeria. Only 5 females of this species were collected in May 2021 on the Sigoise variety at the fruit-setting stage. *Haplothrips tritici*(17.25%), *Frankliniella occidentalis* (16.29%) and *Thrips tabaci* (16.29%) are the most present. It is noticed that the thrips were present on the olive tree only in spring (April to May), when the average monthly temperatures are between 10–26 °C, but linear regression analyses were not confirmed that temperature explain the variation in thrips numbers, which may be due to other climatic factors such as the rainfall, while olive varieties and phenological stages are affecting the population of thrips, their number was higher on the Sigoise variety, especially at flowering stage in the case of *H. tritici* and *F. occidentalis* while *T. tabaci* was most noticeable at the fruit growth stage. The number of this species was relatively low, just until the inflorescence stage, where thrips start to appear in Sigoise before Chemlal.

**Keywords:** olive tree; phytophagous; Thysanoptera

## 1. Introduction

The cultivation of the olive tree (*Olea europea* L.) occupies an important place in the economy of Mediterranean countries, particularly in North Africa, where it plays a very important socio-economic and environmental role [1].unfortunately, throughout the world, this crop is subject to several pests, which negatively affect yields in quantity and quality and annual losses are estimated at more than 15% [2].

Among these pests, phytophagous thrips occupy a very important place. According to Marullo and Vono [3], food bites by can be observed on flower buds and young leaves. On the leaves, these attacks cause necrosis, desiccation, and deformation. On fruit, these stains cause deformation, drying out, and premature drop. These various types of damage lead above all to a reduction in oil yield.

Despite the extent of their damage to olive trees, thrips remain among the least studied groups of insects, particularly in the Maghreb countries. Among the works relating to olive thrips in countries around the Mediterranean, there are especially those of Rei et al. [4] in Portugal, Canale et al. [5] in Italy, Agamy et al. [6] in Egypt, and more recently, Halimi et al. [7] in Algeria. The olive thrips *Liothrips oleae* was already reported in Spain during the nineteen century also in France [8,9], in the Maltese islands [10], and in Italy [3,11], in Algeria it is not yet mentioned. In addition to these records in Portugal, Spain, Greece, and Italy, this thrips is noted more also in Poland, Ethiopia, and Yemen [12,13].

The olive grove in our study consists of two varieties (Chemlal and Sigoise) randomly distributed in the study orchard, of which Chemlal, considered a hardy and late variety,

originating from Kabylia, occupies 40% of the Algerian olive orchard with high productivity intended for the production of oil, with a productivity of 14 to 18 L/quintal [14], on the other hand, Sigoise considered as a seasonal variety, early and tolerant to salt water, their origin is Plane de Sig (Mascara), occupies 25% of the Algerian olive orchard, intended for the production of table olives and oil with average productivity [15].

Polyphagous insects are classified as phytophagous, mycophagous, and predatory, with phytophagous thrips feeding on flowers, fruit, mature leaves, or flower buds [16,17], consequently, so the characteristics and phenological stages of olive varieties can be contributed to their abundant numbers; which can develop in parallel with the different phenological stages of the olive tree; from the hatching of the axillary buds and the appearance of new terminal shoots, in spring; until the flowering as the spring temperature becomes milder, then fruiting and ripening [18].

The principles objective of this study was to evaluate the diversity of thrips associated with olive trees in northeastern Algeria (province of Batna), as the species may be harmful to olive trees, and to study fluctuations in the numbers of thrips adult, according to temperatures and olive varieties phenological stages, during the 2019/2021 period.

## 2. Materials and Methods

### 2.1. Study Site

This study was carried out in an olive grove located in the province of Batna (northeastern Algeria) (6°24′54.72″ E, 35°42′30.24″ N, 875 m) and which is characterized by asemi-arid climate, where cereal growing is the main crop in this region. This olive grove was installed in 2002 and is occupied by 1000 olive trees belonging to the Chemlal and Sigoise varieties. The only standard agronomic practices in the orchard aredeep plowing, drip irrigation system, and tree pruning, without including insecticide application, which is considered an ecological cultivation area. Also, the study orchard is surrounded by a few apple trees and many olive groves.

### 2.2. Diversity of Thrips Communities on Olive Trees

During the period from 2019 to 2021, the olive grove prospected twice a month and the sampling was carried out according to the method proposed by De Borbon [19]. At each survey, the trees were evaluated by randomly, and the canopy lower section of 10 trees per variety (each tree in 4 directions: east, west, north, and south) was beaten by hand on a white plastic tray. Captured specimens were stored in microtubes containing 70% ethanol. According to Reynaud et al. [20], this solution maintains the flexibility of these thrips. Preserved specimens are transferred to NaOH (5%) solution between 2 h (light individuals) and 2 days (dark individuals). Then, the samples were mounted on slides in Hoyer's medium. Voucher specimens were mounted in Canada balsam. Identification of adults was made using the keys provided by zur Strassen and Moritz [21,22]. The main microscopic characters retained for this identification are the number of antennal segments, the shape, and the number of sensory cones, the wing venation, and the number and size of setae on the pronotum [23]. Voucher specimens were deposited in the insect collection of LATPPAM Research Laboratory, University of Batna, and in the National Reference Laboratory for Nematodes and Arthropods, National Museum of Natural Sciences, Madrid.

### 2.3. Statistical Analyzes

Statistical analyses were carried out to determine the variations in the number of thrips species according to varieties and phenological stages. Data were subjected to linear regression analysis, to determine the correlation between average temperatures monthly and thrips abundance, in addition to the study of variance two-way ANOVA test of highly significant differences (HSD) at $p < 0.05$, to analyze the relationship between the thrips numbers and the two independent variables, olives varieties and phenological stages, and the Scheffe test to find out which pairs of means are significant. All analyzes were performed using Microsoft Statistics SPSS version 25 [24].

## 3. Results

### 3.1. Thrips Diversity of on Olive Trees

The results revealed the presence of 19 species of thrips on olive trees during three years of the survey. Among these species, seven species have just been reported for the first time in Algeria among these, *Liothrips oleae* (Figure 1). More than four thrips species were recorded for the first time on olive trees over the world (Table 1), along with, *Liothrips leucopus* (Figure 2). Most thrips identified, were phytophagous (92.01%), and only (7.99%) were facultative predators.

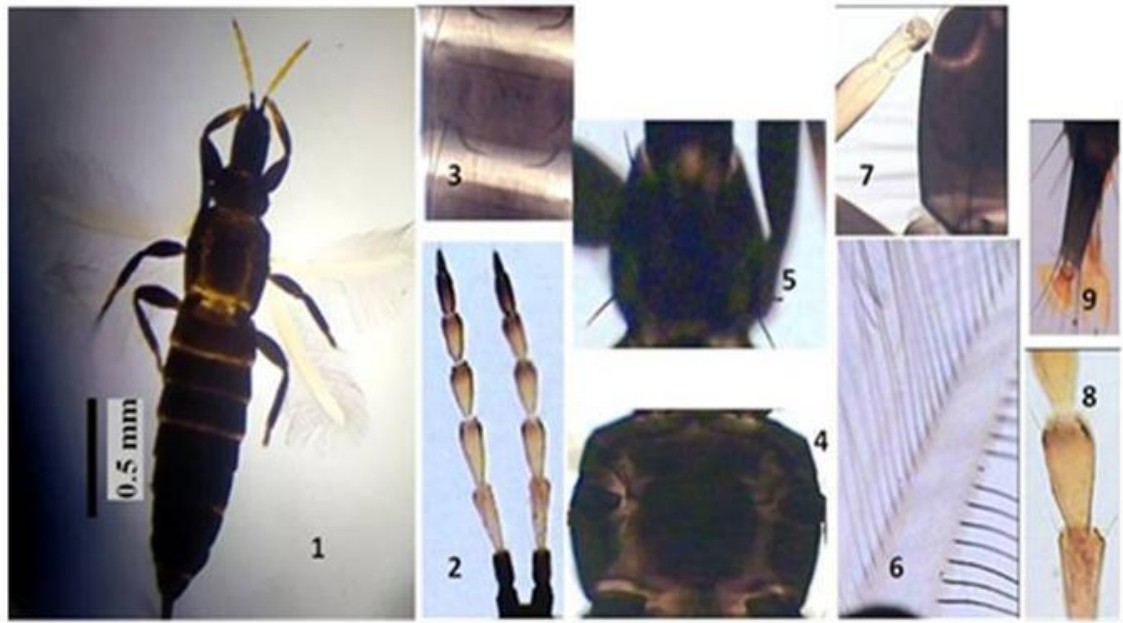

**Figure 1.** 1, *Liothrips oleae* female: 2, antenna; 3, abdominal tergites; 4, mesonotum and metanotum; 5, pronotum; 6, wing; 7, head; 8, antennal segments III–IV; 9, segments IX–X.

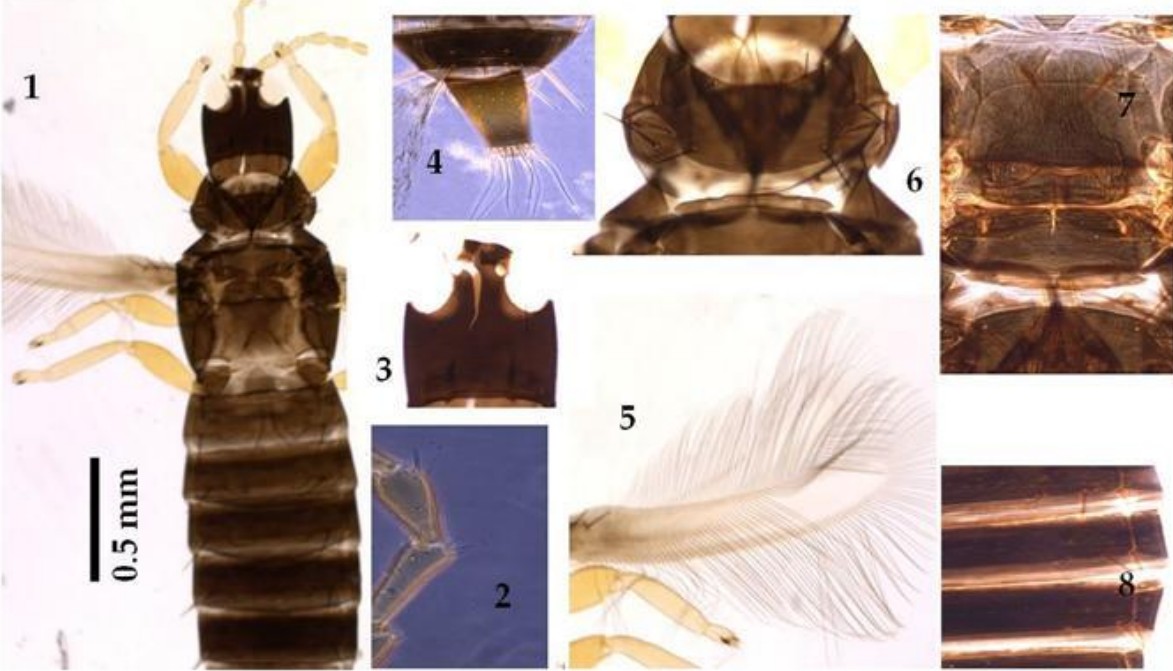

**Figure 2.** 1, *Liothrips leucopus*: 2, antennal segments III–IV; 3, head; 4, segments IX–X; 5, wing pronotum; 6, pronotum; 7, mesonotum and metanotum; 8, abdominal tergites.

**Table 1.** The different species of thrips recorded on olive trees in the province of Batna (2019–2021).

| Suborder | Family | Species | Feed |
|---|---|---|---|
| Terebrantia | Thripidae | *Frankliniella occidentalis* Pergande, 1895 | Phyt. |
| | | *Thrips tabaci* Lindeman, 1889 | Phyt. |
| | | *Thrips angusticeps* Uzel, 1895 | Phyt. |
| | | (*) *Anaphothrips obscurus* Muller, 1776 | Phyt. |
| | | *Thrips minutissimus* Linnaeus, 1758 | Phyt. |
| | | (*) *Stenothrips graminum* Uzel, 1895 | Phyt. |
| | | (*) *Tenothrips frici* Uzel, 1895 | Phyt. |
| | | (*) *Dendrothrips ornatus* Jablonowski, 1894 | Phyt. |
| | Melanthripidae | *Melanthrips fuscus* Sulzer,1776 | Phyt. |
| | Aeolothripidae | (*) *Aeolothrips tenuicornis* Bagnall, 1926 | Fac. Pred. |
| | | *Aeolothrips intermedius* Bagnall, 1934 | Fac. Pred. |
| Tubulifera | Phlaeothripidae | (*) *Liothrips oleae* Costa, 1857 | Phyt. |
| | | (*) (**) *Liothrips leucopus* Titschack, 1958 | Phyt. |
| | | (**) *Dolicholepta micrura* Bagnall, 1914 | Pred. |
| | | *Haplothrips tritici* Kurdjumov, 1912 | Phyt. |
| | | *Haplothrips andresi* Priesner, 1931 | Phyt. |
| | | (**) *Haplothrips crassicornis* John, 1924 | Phyt. |
| | | (**) *Haplothrips distinguendus* Uzel, 1895 | Phyt. |
| | | (*) *Haplothrips aculeatus* Fabricius, 1803 | Phyt. |

(*): First record in Algeria, (**): First record on olive tree, Phyt.: Phytophagous, Fac. Pred.: Facultative predator.

### 3.2. Thrips Adults numbers on Olive Trees

313 specimens of the 19 thrips species were collected in this study; it revealed a variation in the distribution of thrips over the three years of sampling, the majority of them was in 2019 by the maximum number of thrips (155 specimens), followed by 98 and only 60 specimens in 2021 and 2020, respectively, the principal species were *H. tritici, F. occidentalis, T. tabaci,* and *Thrips angusticeps*, accounting for 17.25%, 16.29%, 16.29%, and 10.87% of the total number of thrips, respectively (Figure 3).

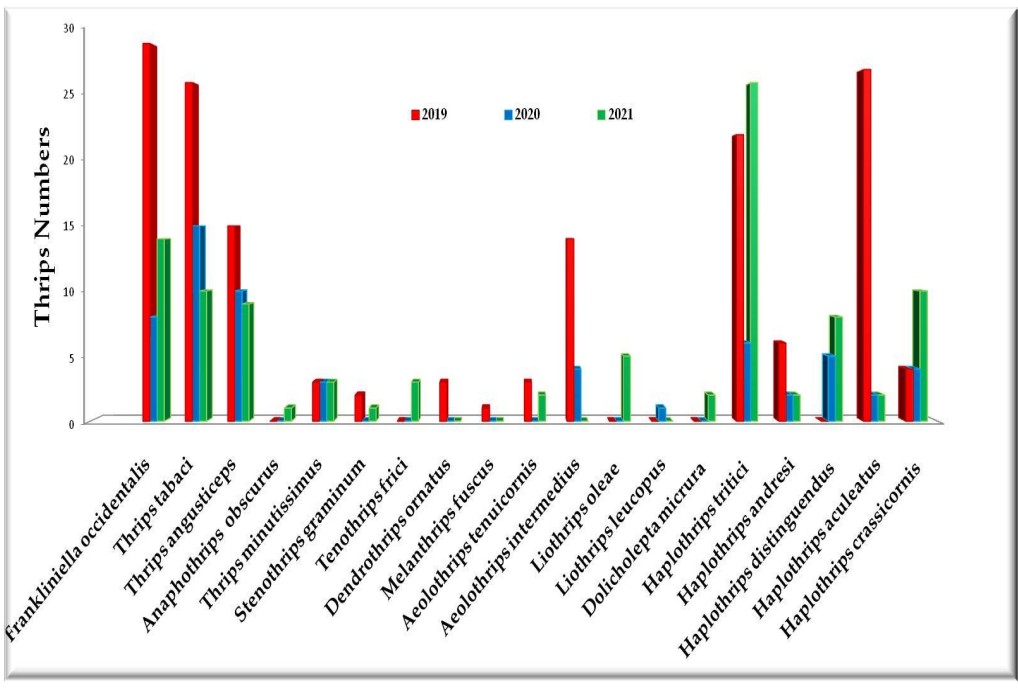

**Figure 3.** Temporal variation of thrips numbers in olive trees in the province of Batna (north-eastern Algeria) during the 2019/2021 period.

### 3.3. Abundance of Thrips Adult Numbers According to the Temperatures

During the period of the research, thrips adults on olive trees have not determined in winter (January and February). while it appears in spring (March–May), where during May they recorded the maximum of his numbers when the average monthly temperatures are between 10–26 °C. But in the summer months, their numbers fall again with the increase in temperatures until they disappear in autumn (Figure 4). Also, we have noticed that April to May 2021 shows anomalous values that are not consistent with the results for 2020 and 2019, this may be due to other climatic factors.

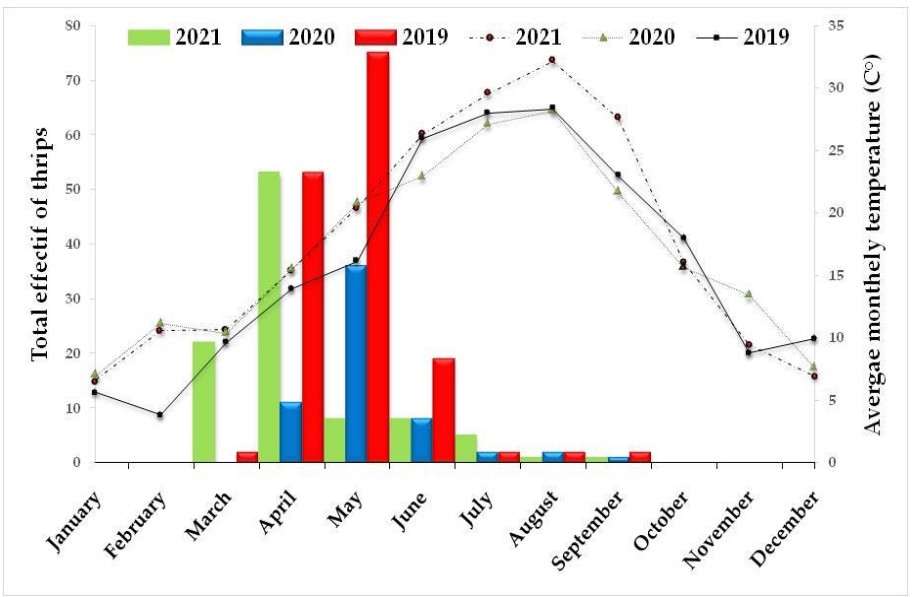

**Figure 4.** Temporal fluctuation of thripsadultson olive trees in Batna region according to monthly average temperatures.

Linear regression analyses (Figure 5) were not significant ($p$ = 0.37) and the temperature doesn't explain a proportion of variation in the rate of thrips numbers, where the value showed a poorly positive correlation between average temperatures and thrips abundance, where the temperature doesn't contribute in the variation in thrips abundance (R-Square = 0.010).

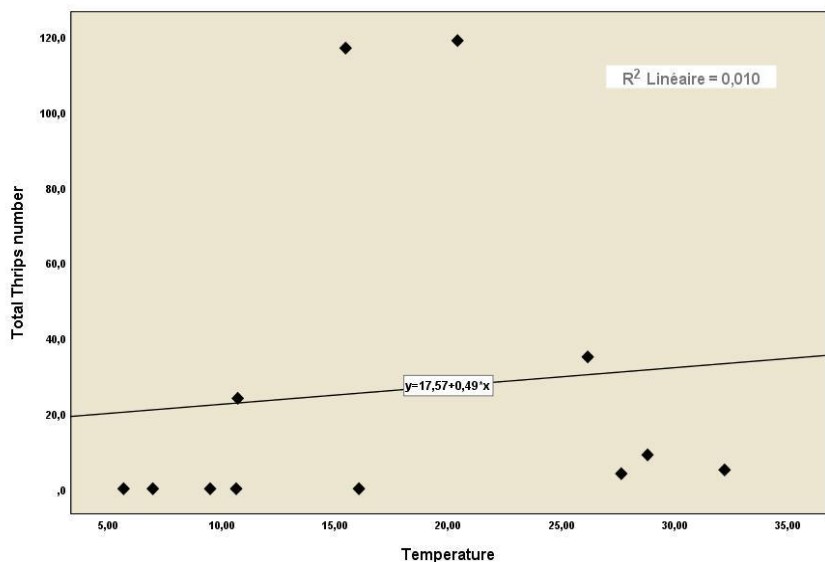

**Figure 5.** Thrips numbers relationship with temperatures average in Batna region.

### 3.4. Distribution of Thrips adults Numbers According to Olive Varieties Phenological Stages

The results obtained prove that on the Sigoise variety thrips numbers were higher than it was on the Chemlal variety. Results of statically testing for the influence of the varieties and phenological stages on the presence and abundance of thrips species showed that there was a significant effect of each factor on the total number of thrips with (F = 19.60, df = 1, $p = 0$), and (F = 11.68, df = 8, $p = 0$) successively, also specifying which stage is favorable for each variety, where the fruit growth (1st stage) was the most attractive for the Sigoise variety, while the average number of this species during the flowering stage is significantly higher than the other phenological stages in the Chemlal variety (Figure 6).

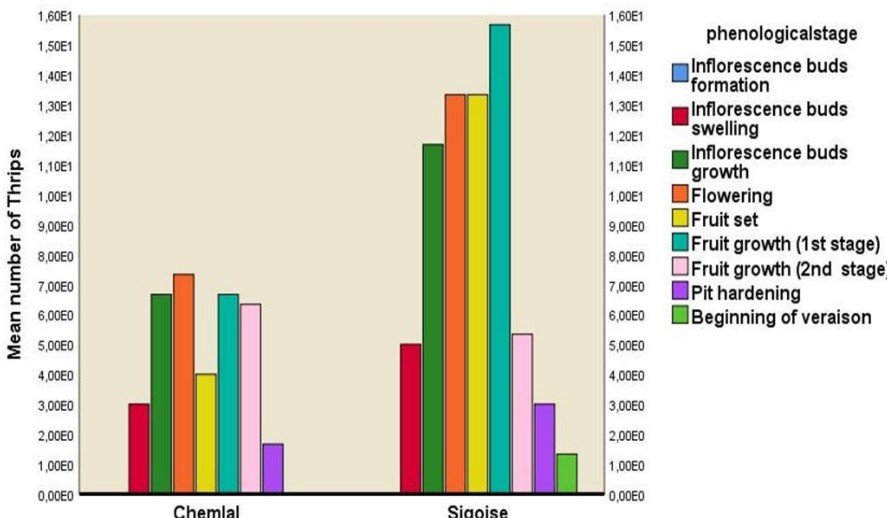

**Figure 6.** Mean number ofThrips adultsaccording to olive varieties phenological stages in Batna region.

### 3.5. The Principal Phytophagous Species Numbers

The abundance of the principal phytophagous species varied according to olive varieties phenological stages, and the activity of these thrips was more important on Sigoise than the Chemlal variety. Scheffe test confirmed the result obtained, with a significant value(F = 9, df = 8, $p = 0$), where the average number of *H. tritici* (3.33 individuals) (Figure 7), during the Flowering stage was significantly higher than the other phenological stages in both varieties. The same result concerning the phenological stage preferably was observed with *F. occidentalis* (3.67 individuals) (Figure 8), where the flowering stagewas the most attractive, while the main number of *T. tabaci* (Figure 9)was more important in the fruit growth stage with(2.66 individuals) and (F = 8.83, df = 8, $p = 0$).

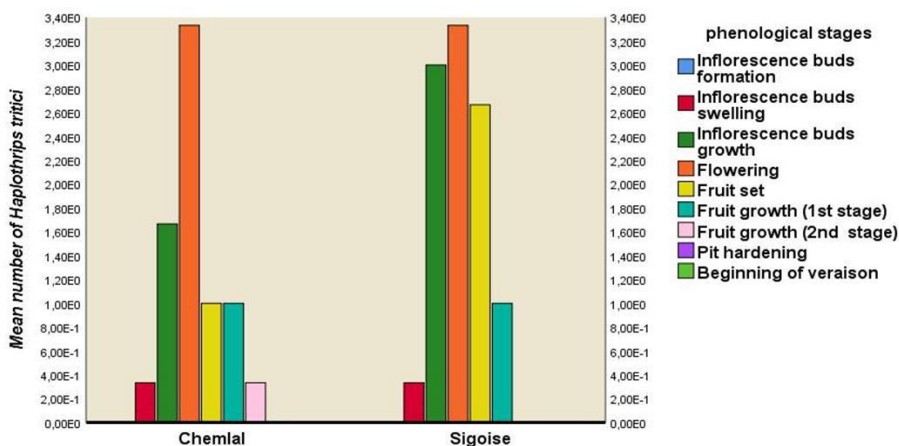

**Figure 7.** Mean number of *H. tritici* according to olive varieties phenological stages in Batna region.

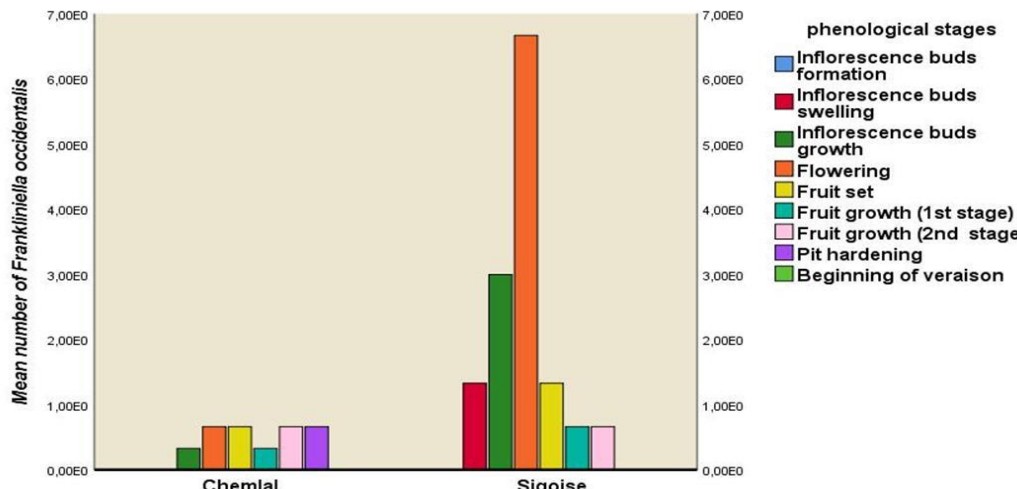

**Figure 8.** Mean number of *F. occidentalis* according to olive varieties phenological stages in Batna region.

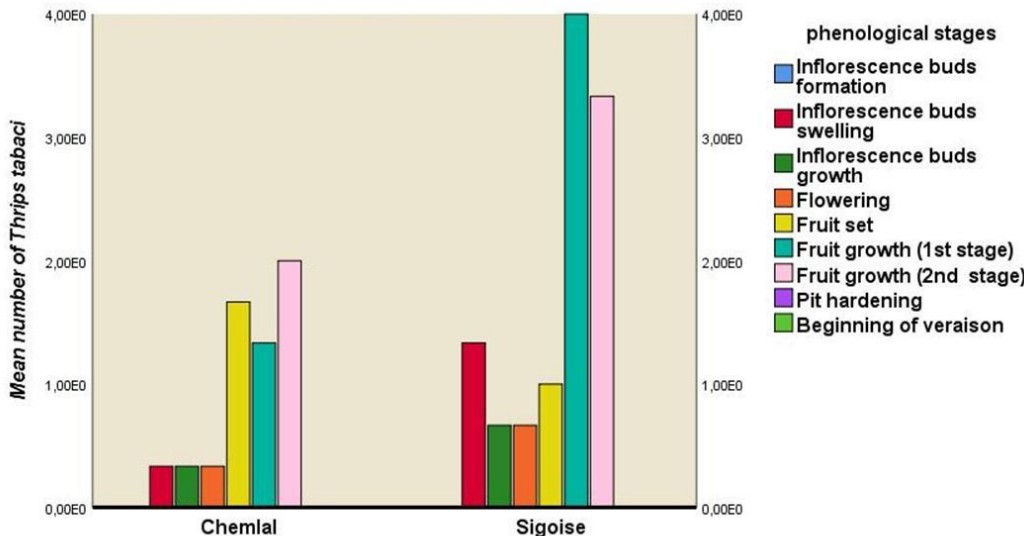

**Figure 9.** Mean number of *T. tabaci* according to olive varieties phenological stages in Batna region.

## 4. Discussion

The majority of thrips species collected on olive trees were phytophagous, more than predators; which may be due to the nature of the study area which is highly agricultural, and the regular agricultures techniques practice, such as irrigation by drop. Some of these species found in the olive trees probably do not reproduce in the olive leaves or fruits. For example, *A. obscurus, S. graminum, H. tritici,* and *H. aculeatus* breeding Gramineae (Poaceae) and *Phragmites australis*. *T. minutissimus* laid eggs on flowers of Quercus in Europe and in Spain and is frequently collected in flowers of Genista, Bellis, and Sinapis [25]. *T. angusticeps* is associated with Cruciferae and *Linum sp.* but it has been collected in Poaceae, Fabaceae, and Compositae in Mediterranean and eurosiberian Spain [25,26]. *M. fuscus* is also abundant in flowers of Brassicaceae which it probably breeds [27]. The presence of these species in the olive leaves and flowers could be possible due to the ecological cultivation of olives in the area of sampling, as no chemical treatment is applied, where the presence of many indigenous plants between the trees is common. *Liothrips leucopus* was originally described from one male collected in *Quercus ilex* in Montpellier (France) [28], however, some populations have been collected posteriorly in Andalusia (Spain) and Marocco by Dr. Richard zur Strassen. Looking carefully at these specimens kindly borrowed by The Senckenberg Museum (Frankfurt am Main, Germany) where are deposited, we

observed a variation in the color of the femora between the populations collected in Spain and North Africa. The specimens collected in Andalusia (Spain) have brown femora in contrast with the North African specimens with yellow femora and tibiae. The Algerian specimen has the leg completely yellow, similar to the Maroccan specimens. Unfortunately, we do not observe more differences between the specimens of both populations to support a specific separation between both populations.

The samples taken in our study revealed greater biodiversity (16 phytophagous species and 3 predatory species), compared with all other works relating to olive thrips, such as Rei et al. [4] in Portugal where they idenitfted9 species of thrips (8 of them are phytophagous), also Canale et al. [5] in Italy has determined 14 phytophagous species and only 2 predatory species, while in Egypt Agamy et al. [6] they represented only 7 phytophagous species, and more recently, Halimi et al. [7] in Algeria with 9 species (7 phytophagous and 2 predatory).

Temperature is one seasonal factor that might impact thrips population dynamics [29,30]. According to Lowry et al. [29], they observed on Peanut, that *Frankliniella fusca* and *F. occidentalis* have an inverse connection between developmental time and temperature, as is typical for all insects. Where the number of degree days required to complete one generation for this thrips reported being 234 (lower threshold 10.5 °C) and 253 (6.5 °C), respectively.

According to Loomans and van Lenteren [31], temperatures between 25 and 30 °C are ideal for the development of thrips, however, during the period of the research, the average monthly temperatures for winter (January and February) were not higher than 5 °C, which may have had a negative impact on thrips development because it would not have been able to resume its activities at these temperatures. While the ambient temperatures observed during May recorded the maximum thrips numbers. In general, thrips were present on the olive tree only in spring (March–May), when the average monthly temperatures are between 10–26 °C. It appears that the May activity is more significant. Additionally, Halimi et al. [7] confirmed that the spring activity of the thrips *Neohydatothrips amygdali* on olive trees is more important, recorded in May (spring) more than in October (autumn).

Also, the anomalous values that are noticed that April to May 2021; which are not consistent with the results for 2020 and 2019, maybe due to other climatic factors, such as the rains, because, when we checked the climatic data, we noticed a difference in the rainfall during these two months, where it is in decreasing (from 53 mm to 29.6 mm in 2019, and from 29.9 mm to 22.5 mm in 2020), while in 2021 it is increasing (from 32.2 mm to 47 mm), According to Bournier [30], the temperature and the hygrometry act in parallel, on the effect in thrips numbers, of which the heavy rains are responsible for the destruction of the majority of their populations.

Actually, the fluctuation of the seasonal thrips population is influenced by various environmental factors including climate, host-plant variety, topography, soil type, and management regimes [32]. According to Etebari et al. [32], the mulberry thrips (*Pseudodendrothrips mori*) show some host preference in their feeding activity, whereas the Kenmuchi variety is preferred by thrips, being that has more proteins which are important nutrients for thrips development in comparison to the other mulberries varieties. Thierry et al. [33] add also that the selection of plants or varieties by phytophagous insects may depend in part on the physical characteristics of feeding and oviposition sites. This includes leaf size, shape, color, thickness, the density of trichomes and stomata, the presence of minerals such as epidermal wax crystals or silica, also the texture and relief of plant surface.

Moreover, as the two varieties are exposed to the same environmental conditions and benefit from the same cultivation techniques, differences in thrips numbers can be only attributed to intrinsic factors, with their morphological differences and the abundance of some primary and secondary metabolites being related. In fact, Mendel & Sebai [15], confirmed the variation in morphological characteristics of each olive variety. They showed that Sigoise is an early maturing variety with medium vigor, long leaves, few flowers, and ovoid fruits, while Chemlal is a late maturing variety with high vigor, average-sized leaves, medium number of flowers, and elongated fruits. Consequently, the early maturity of the variety also contributed to the attractiveness of thrips, because it is noticed that, the Sigoise

variety which was the most preferred, reached its early maturity about two weeks before the Chemlal variety. This was confirmed by the results of Voorrips et al. [34], which show that early plant development leads to a larger thrips population and more severe damage later in the season.

During the period of the early phenological stages of both varieties, the number of this species was relatively low, according to Halimi et al. [7]. This may be due to the absence of flowers and fruit on the olive trees, because just until the inflorescence buds swelling and growth, where thrips start to appear in Sigoise before Chemlal. However, after the bloom period, their numbers increased considerably. They registered their major presence by maximum numbers in April and May in both varieties. This maximum number coincides with the flowering, fruit set, and growth stages. Our results are in agreement with those obtained by Halimi et al. [7], who mentioned that in the olive tree the activity of *N. amygdali* starts only after fruit formation (fruit set), which occurs before the end of April and the beginning of May. Similarly, *F. occidentalis* and *H. tritici* were present in a single peak in April and early May. This peak coincides with the flowering stage.

## 5. Conclusions

The results revealed the presence of 19 species of thrips on olive trees in northeast Algeria (Province of Batna), during three years of the survey (2019–2021), among these species, seven species have just been reported for the first time in Algeria among these, *L. oleae*, and More than four thrips species were recorded for the first time on olive trees over the world, along with, *L. leucopus*. The majority of thrips species collected on olive trees were phytophagous, more than predators.

Although the temperature is one seasonal factor that might impact thrips population dynamics, also we noticed that ambient temperatures observed in spring (March–May) recorded the maximum thrips numbers, when the average monthly temperatures are between 10–26 °C. however, Linear regression analyses were not significant and the temperature doesn't explain a proportion of variation in the rate of thrips numbers, which may be due to other climatic factors; because, according to climatic data during the study months, we noticed an inverse relationship between thrips numbers and the rainfall, in which that heavy rains are responsible for the destruction of the majority of their populations.

While olive varieties and phenological stages are important factors affecting the population dynamics of *H. tritici*, *F. occidentalis*, and *T. tabaci*, the activity of these thrips was more important on Sigoise than the Chemlal variety, where the flowering stage was the most attractive of *H. tritici* and *F. occidentalis*. Despite this stage did not attract *T. tabaci*, the fruit growth stage caused its high abundance.

During the period of the early phenological stages of both varieties, the number of this species was relatively low, which may be due to the absence of flowers and fruit on the olive trees, because just until the inflorescence buds swelling and growth, where thrips start to appear in Sigoise before Chemlal. However, after the bloom period, their numbers increased considerably. They registered their major presence by maximum numbers in April and May in both varieties. This maximum number coincides with the flowering, fruit set, and growth stages.

**Author Contributions:** All authors contributed to the revision of the manuscript. R.M. worked on all experiments in the laboratory, the collection trips, the data analyses, and the writing of the manuscript. M.L. worked on the designed and plan of the study, supervised the experiments, and contributed to writing the manuscript. A.G. worked on the identification of thrips species. All authors have read and agreed to the published version of the manuscript.

**Funding:** This research received no external funding.

**Institutional Review Board Statement:** The insect collections were made between 2019 and 2021 through the certificate issued by University of Batna 1 with the permission to collect specimens with scientific research, Thrips were collected on private property and permission was received from the landowners before sampling procedures.

**Data Availability Statement:** The datasets supporting the conclusions of this article are included within the article and its additional file.

**Acknowledgments:** We thank the owners of the orchard who permitted for the collection of biological material. We thank Andrea Vesmanis (The Senckenberg Museum, Frankfurt) for sending slides to certified *Liothrips leucopus* and Laurence Mound for advice to search the holotype of *L. leucopus*.

**Conflicts of Interest:** The authors declare no conflict of interest.

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
