# Peer review of "Assessment of Thrips Diversity Associated with Two Olive Varieties (Chemlal & Sigoise), in Northeast Algeria"

_horticulturae, doi:10.3390/horticulturae9010107_

Round 1

Reviewer 1 Report

# horticulturae-2124866

The manuscript sought to record the thrips diversity and abundance in olive crops in Northeast Algeria.  From three years of data collection, the authors report 19 species associated with those crops and the first-time detection of plenty of species. The main contribution comes from the species recorded in the area and some association regarding species, crop cultivar, and phenology.

However, the manuscript demands improvement before publication because it requires justification and changes for the authors’ findings, data interpretation, presentation, and analysis.

My comments and suggestion are below:

#Introduction

This section is entirely written in one paragraph, and it is hard to follow the author’s ideas and background presentation. Please consider splitting it up.

Since this is a horticultural journal, I recommend that the crop be the central message instead of pests. For instance, two factors reported in the results – crop variety and phenology are not even mentioned in the introduction. What would we expect in thrips diversity based on them? Not informed nor predictions of the study indicated either. It’s also unclear why the authors focus on one species in lines 33-35, but in lines 80-81, they indicated seven species reported for the first time.

#Materials and methods

L 51-55: Please include the standard agronomic practices in the crops, including insecticide application, if so. Also, provide the surrounding landscape composition since it can be a source of thrips for crops.

L 59-60: Which plant canopy section was beaten (upper, middle, or lower)? What was the (pseudo) replication? Tree or branches? I would say tree, but it’s unclear why the authors primarily indicate the number of branches. How did the authors choose the trees evaluated? Did it vary in each evaluation? Please provide these pieces of information to help readers better understand the sampling method used.

L 71-76: The authors have random (trees, months) and fixed (crop variety and phenological stage) effects in the data, but they are not indicated here or somewhere in the manuscript. Nor how they deal with pseudoreplication in the models. The data used to build the temperature association model also needed to be informed (on the day of collection, 15, 30, or 45 days previously?).

In addition, the authors used ANOVA, followed by Tukey. However, the data shown in figures 3-9 seems unsuitable for ANOVA assumptions due to possible zero inflation, non-residual normality or even homogeneity of variances. Did the authors check it? Likewise, since the same areas were sampled over time, it indicates possible pseudoreplicates that may inflate the model presented. What did the authors include in the model? Please carefully revisit the analyses and provide clear information on which was tested by each model performed.

#Results

The quality of the figures is low, and the axes cannot be adequately seen (e.g., Figure 3). In addition, the authors used the Tukey test after ANOVA, but no letters are indicated to support the author’s claims (e.g., Figures 6-9).

L 120-130: This paragraph seems more discussion rather than results. Please review it to report the results consistently.

L 134-141: Was the regression significant? P-values are not indicated here, and the r-square value is lower for such an important factor, as pointed out in lines 120-130. In addition, the y-axis in figure 5 shows the maximum number of thrips as 5, while figure 3 shows 30 for some species. What is the author’s explanation for this disagreement?

L 144-152: Results not supported by the figures. The degree of freedom informed needs to be clarified. For instance, 23 comes from where? The authors had 36 months of evaluation ( 3 years), so I expected something related. Please revise.

L 156-163: Again, results are not supported by the figures informed. Not clear either why the authors evaluated the three species. It should be indicated why the authors chose to do so.

#Discussion

The authors take to address the main findings of the manuscript succinctly. For instance, which factors do the authors attribute to the ‘high’ number of species found?

The arguments regarding the environmental factors are weak (lines 199-208), as the temperature was the only variable explored and not significantly associated (Figure 5). Plus, some association with the crop is presented in lines 220-231, but there needs to be prior background about the crop varieties to support it.

L 182-186: This information can be previously mentioned in material and methods. Also, the meaning of ecological cultivation needs to be clarified. What do the authors mean?

L 190-194: how does it contribute to the results observed? There is a vague explanation for the results reported.

L 196-198: Phytophagous of predatory species?

L 209-210: Damage was not evaluated in the study. Such an argument is not valid to the results found here.

L 212: Mulberry thrips refers to which species? The first mention of a common name is unclear of the 19 species the authors refer to. 

Author Response

Comments and Suggestions for Authors

# horticulturae-2124866

The manuscript sought to record the thrips diversity and abundance in olive crops in Northeast Algeria.  From three years of data collection, the authors report 19 species associated with those crops and the first-time detection of plenty of species. The main contribution comes from the species recorded in the area and some association regarding species, crop cultivar, and phenology.

However, the manuscript demands improvement before publication because it requires justification and changes for the authors’ findings, data interpretation, presentation, and analysis.

My comments and suggestion are below:

Line numeration was changed after correction

#Introduction

  1. This section is entirely written in one paragraph, and it is hard to follow the author’s ideas and background presentation. Please consider splitting it up.
  2. Answer

I have splitting it up the Introduction in to paragraphs, Line: 25-59.

  1. Since this is a horticultural journal, I recommend that the crop be the central message instead of pests. For instance, two factors reported in the results – crop variety and phenology are not even mentioned in the introduction. What would we expect in thrips diversity based on them? Not informed nor predictions of the study indicated either.
  2. Answer

We rewrote the introduction and add these points; in Line: 25-59.

We expect in thrips diversity, the increase in the number of thrips according to the phenological stages of the olive tree and according to the food source offered to the thrips according to the food regime of each species (feeding on flowers, fruits, or leaves).

  1. it’s also unclear why the authors focus on one species in lines 33-35, but in lines 80-81, they indicated seven species reported for the first time.
  2. Answer

the cause of the focus on one species in lines 33-35, despite the  indication of  seven species reported for the first time; it's in relation of his principle host plants , Some of these species founded in the olive trees probably does not reproduce in the olive leaves or fruits. For example, Anaphothrips obscurus, Stenothrips graminum, Haplothrips aculeatus breedin graminae (Poaceae) and Phragmites australis. The presence of these species in the olive leaves and flowers could be possible due to the presence of many indigenous plants between the trees is common.

 while liothrips oleae it's an insect pest of olive crops and the only host plants is the Olive.

#Materials and methods

  1. L 51-55: Please include the standard agronomic practices in the crops, including insecticide application, if so. Also, provide the surrounding landscape composition since it can be a source of thrips for crops.
  2. Answer

L 62-69: This study was carried out in an olive grove located in the province of Batna (northeastern Algeria) (6°24′54.72″E, 35°42′30.24″N, 875m) and which is characterized by a semi-arid climate, where cereal growing is the main crop in this region. This olive grove was installed in 2002 and is occupied by 1000 olive trees belonging to the Chemlal and Sigoise varieties. The only standard agronomic practices in the orchard are deep plowing, drip irrigation system, and tree pruning, without including insecticide application, which is considered an ecological cultivation area. Also, the study orchard is surrounded by a few apple trees and many olive groves.

  1. L 59-60: Which plant canopy section was beaten (upper, middle, or lower)? What was the (pseudo) replication? Tree or branches? I would say tree, but it’s unclear why the authors primarily indicate the number of branches. How did the authors choose the trees evaluated? Did it vary in each evaluation? Please provide these pieces of information to help readers better understand the sampling method used.
  2. Answer

L 72-75: At each survey, the trees were evaluated by randomly, and the canopy lower section of 10 trees per variety (each tree in 4 directions: east, west, north, and south) was beaten by hand on a white plastic tray.

  1. L 71-76: The authors have random (trees, months) and fixed (crop variety and phenological stage) effects in the data, but they are not indicated here or somewhere in the manuscript. Nor how they deal with pseudoreplication in the models. The data used to build the temperature association model also needed to be informed (on the day of collection, 15, 30, or 45 days previously?).
  2. Answer

L 88-94: Statistical analyzes were carried out to determine the variations in the numbers of thrips species according to varieties and phenological stages. Data were subjected to linear regression analysis, to determine the correlation between average temperatures monthly and thrips abundance. In addition to the analysis of variance two-way ANOVA test of highly significant differences (HSD) at P<0.05, was followed by the Scheffe test to find out which pairs of means are significant. All analyzes were performed using Microsoft Statistics SPSS version 25 [24].

The ways how we deal with pseudoreplication in the models  are: average the pseudoreplicates to obtain one value per genuine replicate,The data used to build the temperature association model on the average temperature monthly of capture.

  1. In addition, the authors used ANOVA, followed by Tukey. However, the data shown in figures 3-9 seems unsuitable for ANOVA assumptions due to possible zero inflation, non-residual normality or even homogeneity of variances. Did the authors check it? Likewise, since the same areas were sampled over time, it indicates possible pseudoreplicates that may inflate the model presented. What did the authors include in the model? Please carefully revisit the analyses and provide clear information on which was tested by each model performed.
  2. Answer

Figures 3-4 don't undergo ANOVA assumptions because it schematized by Excel, but the other Figures 5-9; were redone by the analysis of variance two-way ANOVA test, and revisited the analyses.

#Results

  1. The quality of the figures is low, and the axes cannot be adequately seen (e.g., Figure 3). In addition, the authors used the Tukey test after ANOVA, but no letters are indicated to support the author’s claims (e.g., Figures 6-9).
  2. Answer

Figures; redone and (e.g., Figure 3) the axes can be adequately seen ,we indicated the Scheffe test.

  1. L 120-130: This paragraph seems more discussion rather than results. Please review it to report the results consistently.
  2. Answer

Line 144-149: I added it to the discussion, and the results are as follows:

During the period of the research, thrips adults on olive trees have not been able to resume their activities in winter (January and February). And it appears in spring (March-May), where during May they recorded the maximum of his numbers when the average monthly temperatures are between 10-26°C. But in the summer months, their numbers fall again with the increase in temperatures until they disappear in autumn (figure 4).

  1. L 134-141: Was the regression significant? P-values are not indicated here, and the r-square value is lower for such an important factor, as pointed out in lines 120-130. In addition, the y-axis in figure 5 shows the maximum number of thrips as 5, while figure 3 shows 30 for some species. What is the author’s explanation for this disagreement?
  2. Answer

The regression was not significant (P=0. 37):

lines 153-161: Linear regression analyses (Figure 5) were not significant (P=0.37) and temperature explained a low proportion of variation in the rate of thrips numbers, where the value showed a poorly positive correlation between average temperatures and thrips abundance, where temperature contributed only by 1% in the variation in thrips abundance (R-square= 0.010). This correlation confirms the results in (Figure 4), which shows that thrips insects completely disappear when average temperatures are below 5°C; it may be due to the semi-arid environment, which is dry and cold in winter and very hot in summer in the Batna region, which records very low temperatures in January and February, explains the low value of (R-square).

the y-axis (we fixed it) in figure 5 shows the maximum number of thrips as 120, while figure 3 shows 30 for some species. the explanation for this disagreement; is that figure 3 shows the thrips number for each species; while Figure 5 shows the total thrips number in each month.

  1. L 144-152: Results not supported by the figures. The degree of freedom informed needs to be clarified. For instance, 23 comes from where? The authors had 36 months of evaluation ( 3 years), so I expected something related. Please revise.
  2. Answer

We revise it, and changed the analysis to a two-way ANOVA:

L 165-172: The results obtained prove that on the Sigoise variety thrips numbers were higher than it was on the Chemlal variety. Results of statically testing for the influence of the varieties and phenological stages on the presence and abundance of thrips species showed that there was a significant effect of each factor on the total number of thrips with (F=19.60, df= 36, P=0), and (F=11.68,df= 36, P=0) successively, also specifying which stage is favorable for each variety, where the fruit growth (1st stage) was the most attractive for the Sigoise variety, while the average number of this species during the flowering stage is  significantly higher than the other phenological stages in the Chemlal variety (Figure 6).

  1. L 156-163: Again, results are not supported by the figures informed. Not clear either why the authors evaluated the three species. It should be indicated why the authors chose to do so.
  2. Answer

the authors evaluated the three species; because are the most abundant in number compared to other determined species, where it had been saying before in L 137-139: , respectively, the principal species were H. tritici, F. occidentalis, T. tabaci, and Thrips angusticeps, accounting for 17.25%, 16.29%, 16.29%, and 10.87% of the total number of thrips, respectively (figure 3).

L 186-194: The abundance of the principal phytophagous species varied according to olive varieties phenological stages, and the activity of these thrips was more important on Sigoise than the Chemlal variety. Scheffe test confirmed the result obtained, with a significant value( P≤ 0, df= 36), where the average number of H. tritici (3.33 individuals) (Figure 7), during the Flowering stage was significantly higher than the other phenological stages in both varieties. The same result concerning the phenological stage preferably was observed with F. occidentalis (3.67 individuals) (Figure 8), where the flowering stage was the most attractive, while the main number of T. tabaci(Figure 9)was more important in the fruit growth stage(2.66 individuals).

#Discussion

  1. The authors take to address the main findings of the manuscript succinctly. For instance, which factors do the authors attribute to the ‘high’ number of species found?
  2. Answer

This "high" number of species found may be related to the non-use of pesticides in the treatment of trees at the study orchard.

  1. The arguments regarding the environmental factors are weak (lines 199-208), as the temperature was the only variable explored and not significantly associated (Figure 5). Plus, some association with the crop is presented in lines 220-231, but there needs to be prior background about the crop varieties to support it.
  2. Answer

The arguments regarding environmental factors are weak, due to the lack of articles and works concerning it, we added background about the crop varieties in the introduction,(lines 43-49): The study olive grove consists of two varieties (Chemlal and Sigoise) randomly distributed in the study orchard, of which Chemlal, considered a hardy and late variety, originating from Kabylia, occupies 40% of the Algerian olive orchard with high productivity intended for the production of oil, with a productivity of 14 to 18 liters/quintal [14]  , on the other hand, Sigoise considered as a seasonal variety, early and tolerant to salt water, their origin is Plane de Sig (Mascara), occupies 25% of the Algerian olive orchard, intended for the production of table olives and oil with average productivity [15].

  1. L 182-186: This information can be previously mentioned in material and methods. Also, the meaning of ecological cultivation needs to be clarified. What do the authors mean?
  2. Answer

We mentioned the information in the Study site (L62-69): This study was carried out in an olive grove located in the province of Batna (northeastern Algeria) (6°24′54.72″E, 35°42′30.24″N, 875m) and which is characterized by a semi-arid climate, where cereal growing is the main crop in this region. This olive grove was installed in 2002 and is occupied by 1000 olive trees belonging to the Chemlal and Sigoise varieties. The only standard agronomic practices in the orchard are deep plowing, drip irrigation system, and tree pruning, without including insecticide application, which is considered an ecological cultivation area. Also, the study orchard is surrounded by a few apple trees and many olive groves.

Ecological cultivation means:  that no chemical treatment is applied

L 213-216: The presence of these species in the olive leaves and flowers could be possible due to the ecological cultivation of olives in the area of sampling, as no chemical treatment is applied, where the presence of many indigenous plants between the trees is common.

  1. L 190-194: how does it contribute to the results observed? There is a vague explanation for the results reported.
  2. Answer

L 203-208: The majority of thrips species collected on olive trees were phytophagous, more than predators; which may be due to the nature of the study area which is highly agricultural, and the regular agricultures techniques practice, such as irrigation by drop. Some of these species found in the olive trees probably do not reproduce in the olive leaves or fruits. For example, Anaphothrips obscurus, Stenothrips graminum, Haplothrips tritici, and Haplothrips aculeatus breeding Gramineae (Poaceae) and Phragmites australis.

  1. L 196-198: Phytophagous of predatory species?
  2. Answer

L 227-233: The samples taken in our study revealed greater biodiversity (16 phytophagous species and 3 predatory species), compared with all other works relating to olive thrips, such as Rei et al. [4] in Portugal where they identifted9 species of thrips ( 8 of them are phytophagous), also Canale et al. [5]  in Italy has determined 14 phytophagous species and only 2 predatory species,  while in Egypt Agamy et al. [6] they represented only 7 phytophagous species, and more recently, Halimi et al. [7] in Algeria with 9 species (7 phytophagous and 2 predatory ).

  1. L 209-210: Damage was not evaluated in the study. Such an argument is not valid to the results found here.
  2. Answer

L 209-210: I have eliminated the degree of damage, but this argument is still valid to the results of the fluctuation of seasonal thrips population.

  1. L 212: Mulberry thrips refers to which species? The first mention of a common name is unclear of the 19 species the authors refer to. 
  2. Answer

Mulberry thrips refers to species Pseudodendrothrips mori

L 253: According to Etebari et al. [32], the mulberry thrips (Pseudodendrothrips mori) show some host preference in their feeding activity,

The 19 species the authors refer to their scientific names not common name.

Submission Date :12 December 2022

Date of this review :16 Dec 2022 01:15:32

Reviewer 2 Report

The manuscript titled “Assessment of Thrips Diversity Associated with Two Olive Varieties (Chemlal & Sigoise) , in Northeast Algeria”, which evaluate thrips diversity in two olive varieties (Chemlal & Sigoise) in north-eastern Algeria and the occurrence of important phytophagous thrips during different phenological periods. This study will provide a theoretical basis for the selection of insect-resistant olive varieties. It will be of practical importance for predicting and controlling thrips occurrence by analyzing its relationship with the variability of thrips populations concerning the phenology of olive trees.

 The writing and experimental design of the manuscript is relatively good. However, the language of some paragraphs is not sufficiently concise and logical, and the experiment's purpose and significance are unclear. The discussion lacks a comparison with the results of this study and is not sufficiently in-depth. I recommend that this study be published after a major revision of the manuscript.

Author Response

General comment

The manuscript titled “Assessment of Thrips Diversity Associated with Two Olive Varieties (Chemlal & Sigoise) , in Northeast Algeria”, whichevaluate thrips diversity in two olive varieties (Chemlal & Sigoise) in north-eastern Algeria and the occurrence of important phytophagous thrips during different phenological periods. This study will provide a theoretical basis for the selection of insect-resistant olive varieties. It will be of practical importance for predicting and controllingthrips occurrence by analyzing its relationship with the variability of thrips populations concerning the phenology of olive trees.

The writing and experimental design of the manuscript is relatively good. However, the language of some paragraphs is not sufficiently concise and logical, and the experiment's purpose and significance are unclear. The discussion lacks a comparison with the results of this study and is not sufficiently in-depth. I recommend that this study be published after a major revision of the manuscript.

Comments

Line numeration was changed after correction

  1. Line 120-130 is the writing form of discussion
  2. Answer

Line 144-149: I added it to the discussion, and the results are as follows:

During the period of the research, thrips adults on olive trees have not been able to resume their activities in winter (January and February). And it appears in spring (March-May), where during May they recorded the maximum of his numbers when the average monthly temperatures are between 10-26°C. But in the summer months, their numbers fall again with the increase in temperatures until they disappear in autumn (figure 4).

  1. Figure 6-9 can be combined into a picture
  2. Answer

Figures 6-9 can’t be combined into a picture, because Figure 6: represented the total Mean number of thrips, while Figure 9: represents only the Mean number of Thrips tabaci only.

  1. The age of the morphological identification is indicated (as either a worm or an adult)
  2. Answer

The age of the morphological identification is indicated as an adult.

Line 79-80 : Identification of adults was made using the keys provided by zur Strassen and Moritz[21,22].

  1. The foreword language is not refined enough, and needs to be polished, and the logic should be strengthened, and the purpose and significance of the test should be clearly indicated.
  2. Answer

We rewrote the introduction and add these points; in Line 25-59:

The purpose ; Line 56-59: The principles objective of this study was to evaluate the diversity of thrips associated with olive trees in northeastern Algeria (province of Batna), as the species may be harmful to olive trees, and to study fluctuations in the numbers of thrips adult, according to temperatures and olive varieties phenological stages, during the 2019/2021 period.

  1. The data analysis section should be a two-way ANOVA, and the writing of the regression analysis are missing in data section.
  2. Answer

We changed the analysis to a two-way ANOVA:

Line 88-94: Statistical analyzes were carried out to determine the variations in the numbers of thrips species according to varieties and phenological stages. Data were subjected to linear regression analysis, to determine the correlation between average temperatures monthly and thrips abundance. In addition to the analysis of variance two-way ANOVA test of highly significant differences (HSD) at P<0.05, was followed by the Scheffe test to find out which pairs of means are significant. All analyzes were performed using Microsoft Statistics SPSS version 25 [24].

  1. In the discussion part, there is little mention of your test results, and only a simple comparison of others' results.
  2. Answer

In the discussion part, we mentioned all the results which we obtained. Also which explains the fewer numbers of comparisons of the results with others, that thrips remain among the least studied groups of insects, particularly the works relating to olive thrips, where we cited all works that were released in:  Line 227-233: The samples taken in our study revealed greater biodiversity (16 phytophagous species and 3 predatory species), compared with all other works relating to olive thrips, such as Rei et al. [4] in Portugal where they identified 9 species of thrips ( 8 of them are phytophagous), also Canale et al. [5]  in Italy has determined 14 phytophagous species and only 2 predatory species,  while in Egypt Agamy et al. [6] they represented only 7 phytophagous species, and more recently, Halimi et al. [7] in Algeria with 9 species (7 phytophagous and 2 predatory ).

Submission Date :12 December 2022

Date of this review :23 Dec 2022 10:28:21

Round 2

Reviewer 1 Report

# horticulturae-2124866

The authors provide a revised version of the manuscript. The changes have improved the manuscript sections, especially the introduction. However, the statistical analysis still needs improvements before the manuscript's publication. In addition, the data does not support the author’s claims regarding temperature. Some speculations can be indicated but not assumed as a true/significant effect observed.  Also, the conclusion is not presented in the abstract or end of the discussion.

 #Minor points

Line 11: Class and order do not need to be italicised.

Line 21: The conclusion needs to be included in the abstract.

Line 39-42: The authors maybe consider providing arguments on why this species is of particular importance. Here, they only described where this species had been detected. Is it an invasive species?

Line 43: Unclear. Which study do the authors refer to?

Lines 88-94: The authors may indicate that the assessments were averaged to avoid pseudoreplication in the analysis and if the assumptions were tested and reached during that. When mentioning two-way, which independent variables do the authors refer to?

Lines 144-145: The authors only evaluate thrips’ presence. The mention of ‘activities’ is speculative and unrelated to presence/absence.

Lines 156-159: It does not. The temperature was not significant, and the authors still argue that.

Lines 168 and 167: The degrees of freedom seem incorrect – 3 years (36 months). So, what is the residual degree of freedom?

Line 188: Statistical reports still need to be completed.

Lines 208-209: Species names can be abbreviated since they were reported previously in the manuscript.

Line 283: Conclusion?

Author Response

Comments and Suggestions for Authors

# horticulturae-2124866

The authors provide a revised version of the manuscript. The changes have improved the manuscript sections, especially the introduction. However, the statistical analysis still needs improvements before the manuscript's publication. In addition, the data does not support the author’s claims regarding temperature. Some speculations can be indicated but not assumed as a true/significant effect observed.  Also, the conclusion is not presented in the abstract or end of the discussion.

 #Minor points

  1. Line 11: Class and order do not need to be italicised.
  2. Answer

          Line 11: Class and order (Insecta: Thysanoptera).

  1. Line 21: The conclusion needs to be included in the abstract.
  2. Answer

The conclusion needs has included in the abstract; also the Abstract should be a single paragraph of about 200 words maximum;

Line 18-26 : It is noticed that the thrips were present on the olive tree only in spring (April to May), when the average monthly temperatures are between 10-26°C ,but linear regression analyses were not confirmed that temperature explain the variation in thrips numbers, which may be due to other climatic factors such as the rainfall,while olive varieties and phenological stages are affecting the population of thrips,their number was higher on the Sigoise variety, especially at flowering stage in the case of H. tritici and F. occidentalis while T.tabaci was most noticeable at the fruit growth stage. The number of this species was relatively low, just until the inflorescence stage, where thrips start to appear in Sigoise before Chemlal.

  1. Line 39-42: The authors maybe consider providing arguments on why this species is of particular importance. Here, they only described where this species had been detected. Is it an invasive species?
  2. Answer

Liothrips oleae ,is not an invasive species;Line 48-57:     In the spring, the females regain their activity and begin to lay a week later (up to 300 eggs). Immediately after formation, the larvae feed on buds and young leaves before pupating two weeks later [14]. According to this author, this thrips develops between three to four generations per year and its numbers reach their peak towards the end of June and the beginning of July. In Spain, there were important attacks with high economic losses during the eighteen century but the treatments with insecticides seem to reduce the population of Liothrips oleae in the orchards during the XX century (Lacasa, personal communication). The heavy infestations of this species in 2017 in Italy caused significant damage, in particular deformation of young shoots and premature fruit drop [3].

  1. Line 43: Unclear. Which study do the authors refer to?
  2. Answer

     Line 58: The olive grove in our study consists of two varieties (Chemlal and Sigoise).

  1. Lines 88-94: The authors may indicate that the assessments were averaged to avoid pseudo replication in the analysis and if the assumptions were tested and reached during that. When mentioning two-way, which independent variables do the authors refer to?
  2. Answer

Lines 109-113: in addition to the study of variance two-way ANOVA test of highly significant differences (HSD) at P<0.05, to analyze the relationship between the thrips numbers and the two independent variables, olives varieties and phenological stages, and the Scheffe test to find out which pairs of means are significant. All analyzes were performed using Microsoft Statistics SPSS version 25 [24].

  1. Lines 144-145: The authors only evaluate thrips’ presence. The mention of ‘activities’ is speculative and unrelated to presence/absence.
  2. Answer

      Lines 144-145: During the period of the research, thrips adults on olive trees have not determined      in winter (January and February).

  1. Lines 156-159: It does not. The temperature was not significant, and the authors still argue that.
  2. Answer

Lines 156-159: the temperature doesn't explain a  proportion of variation in the rate of thrips numbers, where the value showed a poorly positive correlation between average temperatures and thrips abundance, where the temperature doesn't contribute in the variation in thrips abundance (R-square= 0.010).

  1. Lines 166 and 167: The degrees of freedom seem incorrect – 3 years (36 months). So, what is the residual degree of freedom?
  2. Answer

    We corrected the degrees of freedom to (F=19.60, df= 1, P=0), and (F=11.68,df= 8, P=0)

  1. Line 188: Statistical reports still need to be completed.
  2. Answer

Line 179-184: Scheffe test confirmed the result obtained, with a significant value( F=9,df= 8, P=0), where the average number of H. tritici(3.33 individuals) (Figure 7), during the Flowering stage was significantly higher than the other phenological stages in both varieties. The same result concerning the phenological stage preferably was observed with F. occidentalis (3.67 individuals) (Figure 8), where the flowering stage was the most attractive, while the main number of T.tabaci(Figure 9)was more important in the fruit growth stage with(2.66 individuals) and (F=8.83,df= 8, P=0).

  1. Lines 208-209: Species names can be abbreviated since they were reported previously in the manuscript.
  2. Answer

Lines 197-209:  For example, A. obscurus, S. graminum, H. tritici, and H. aculeatus breeding Gramineae (Poaceae) and Phragmites australis. T. minutissimus laid eggs on flowers of Quercus in Europe and in Spain and is frequently collected in flowers of Genista, Bellis, and Sinapis[25]. T. angusticeps is associated with Cruciferae and Linum sp. but it has been collected in Poaceae, Fabaceae, and Compositae in Mediterranean and eurosiberian Spain [26,25]. M. fuscus is also abundant in flowers of Brassicaceae which it probably breeds[27].

  1. Line 283: Conclusion?
  2. Answer

Conclusions; Lines 286-312: The results revealed the presence of 19 species of thrips on olive trees in northeast Algeria (Province of Batna), during three years of the survey (2019 -2021), among these species, seven species have just been reported for the first time in Algeria among these, L. oleae , and More than four thrips species were recorded for the first time on olive trees over the world, along with, L. leucopus . The majority of thrips species collected on olive trees were phytophagous, more than predators.

  Although the temperature is one seasonal factor that might impact thrips population dynamics, also we noticed that ambient temperatures observed in spring (March-May) recorded the maximum thrips numbers, when the average monthly temperatures are between 10-26°C. however, Linear regression analyses were not significant and the temperature doesn't explain a  proportion of variation in the rate of thrips numbers, which may be due to other climatic factors; because, according to climatic data during the study months, we noticed an inverse relationship between thrips numbers and the rainfall, in which that heavy rains are responsible for the destruction of the majority of their populations.

   While olive varieties and phenological stages are important factors affecting the population dynamics of H. tritici, F. occidentalis,and T.tabaci, the activity of these thrips was more important on Sigoise than the Chemlal variety, where the flowering stage was the most attractive of H. tritici and F. occidentalis. Despite this stage did not attract T.tabaci, the fruit growth stage caused its high abundance.

   During the period of the early phenological stages of both varieties, the number of this species was relatively low, which may be due to the absence of flowers and fruit on the olive trees, because just until the inflorescence buds swelling and growth, where thrips start to appear in Sigoise before Chemlal. However, after the bloom period, their numbers increased considerably. They registered their major presence by maximum numbers in April and May in both varieties. This maximum number coincides with the flowering, fruit set, and growth stages.

Reviewer 2 Report

The manuscript titled “Assessment of Thrips Diversity Associated with Two Olive Varieties (Chemlal & Sigoise) , in Northeast Algeria”, which evaluate thrips diversity in two olive varieties (Chemlal & Sigoise) in north-eastern Algeria and the occurrence of important phytophagous thrips during different phenological periods. This study will provide a theoretical basis for the selection of insect-resistant olive varieties. It will be of practical importance for predicting and controlling thrips occurrence by analyzing its relationship with the variability of thrips populations concerning the phenology of olive trees.

The manuscript is relatively well written and structured after the initial major revision, despite some minor editing problems. I recommend publication of this study after a minor revision of the manuscript.

Author Response

General comment

The manuscript titled “Assessment of Thrips Diversity Associated with Two Olive Varieties (Chemlal & Sigoise) , in Northeast Algeria”, which evaluate thrips diversity in two olive varieties (Chemlal & Sigoise) in north-eastern Algeria and the occurrence of important phytophagous thrips during different phenological periods. This study will provide a theoretical basis for the selection of insect-resistant olive varieties. It will be of practical importance for predicting and controlling thrips occurrence by analyzing its relationship with the variability of thrips populations concerning the phenology of olive trees.

The manuscript is relatively well written and structured after the initial major revision, despite some minor editing problems. I recommend publication of this study after a minor revision of the manuscript.

Comments

  1. Line 30-34 Damage caused by thrips, preferably with practical examples to highlight the importance of phytophagous thrips.
  2. Answer

Line 35-39: Among these pests, phytophagous thrips occupy a very important place. According to Marullo and Vono [3], food bites by olive thrips (Liothrips Oleae) can be observed on flower buds and young leaves. On the leaves, these attacks cause necrosis, desiccation, and deformation. On fruit, these stains cause deformation, drying out, and premature drop. These various types of damage lead above all to a reduction in oil yield.

  1. .Line 50-55 This paragraph is proposed to be reorganized, Polyphagous insects are classified as phytophagous, mycophagous and predatory, with phytophagous thrips feeding on flowers, fruit, mature leaves or flower buds.
  2. Answer

       paragraph  reorganized; Line 66-72: Polyphagous insects are classified as phytophagous, mycophagous, and predatory, with phytophagous thrips feeding on flowers, fruit, mature leaves, or flower buds [16,17], consequently, so the characteristics and phenological stages of olive varieties can be contributed to their abundant numbers; which can develop in parallel with the different phenological stages of the olive tree; from the hatching of the axillary buds and the appearance of new terminal shoots, in spring; until the flowering as the spring temperature becomes milder, then fruiting and ripening  [18].

  1. Line 144-148 April to May 2021 show anomalous values that are not consistent with the results for 2020 and 2019, consider the reasons for this.
  2. Answer

Line 145-151: During the period of the research, thrips adults on olive trees have not determined in winter (January and February). while it appears in spring (March-May), where during May they recorded the maximum of his numbers when the average monthly temperatures are between 10-26°C. But in the summer months, their numbers fall again with the increase in temperatures until they disappear in autumn (figure 4). Also, we have noticed that April to May 2021 shows anomalous values that are not consistent with the results for 2020 and 2019, this may be due to other climatic factors.

the reasons for this; Line 241-248: Also, the anomalous values that are noticed that April to May 2021; which are not consistent with the results for 2020 and 2019, may be due to other climatic factors, such as the rains, because, when we checked the climatic data, we noticed a difference in the rainfall during these two months, where it is in decreasing (from 53mm to 29.6mm in 2019, and from 29.9mm to 22.5mm in 2020), while in 2021 it is increasing (from 32.2mm to 47mm), According to Bournier[29], the temperature and the hygrometry act in parallel, on the effect in thrips numbers, of which the heavy rains are responsible for the destruction of the majority of their populations.

  1. The spacing of the last paragraphs of the Discussion section is not consistent.
  2. Answer

The spacing of the last paragraphs of the Discussion section is consistent now, after reforming
